# The History of Early Breast Cancer Treatment

**DOI:** 10.3390/genes13060960

**Published:** 2022-05-27

**Authors:** Judith Ben-Dror, Michal Shalamov, Amir Sonnenblick

**Affiliations:** 1Tel Aviv Sourasky Medical Center, Tel Aviv 6423906, Israel; 2Sackler Faculty of Medicine, Tel Aviv University, Tel Aviv 6997801, Israel

**Keywords:** breast cancer, molecular therapeutics, systemic treatment, targeted therapies

## Abstract

“The story of cancer is the story of human ingenuity, resilience, and perseverance, but also of hubris, paternalism, and misperception” (Siddhartha Mukherjee). The present review discusses the evolution of early breast cancer (BC) treatment philosophy in the last 50 years and the shift from an emphasis on local therapy to an emphasis on systemic precision treatment options.

## 1. Introduction

Breast cancer (BC) is the most common cancer among women. About 42,000 women die of BC every year, making it the second most common cancer and the fifth leading cause of death worldwide [1]. In 2020, nearly 2.3 million women were diagnosed with BC and 685,000 deaths resulted from the disease globally. BC occurs in women of all ages worldwide, however at increasing rates later in life.

BC has been diagnosed since ancient times. Hippocrates, “the father of Western medicine”, first suggested that BC was a systemic disease. The fundamentals of his theory stood for more than 2000 years until 1757 when Henri Le Dran, a French physician, suggested that the surgical removal of the tumor could cure BC as it was a local disease with local origin. However, it was not until the end of the nineteenth century when William Halsted reinvigorated Le Dran’s theory and performed the first radical mastectomy BC [2]. The Halsted procedure involved the removal of the entire breast, axillary nodes and chest muscles. “Halsted mastectomy” became the gold standard for more than half a century, leaving many women with major disabilities.

## 2. Local Treatment

### 2.1. Surgery

The National Surgical Adjuvant Breast and Bowel Project (NSABP) originated in 1957 to test the effectiveness of various anticancer treatments used with cancer surgery. It outlined the fundamentals of clinical trials, which included a predefined protocol, specific inclusion and exclusion criteria, and adherence to strict randomization procedures. In 1967, Bernard Fisher was appointed the chairman of the NSABP and, a few years later, he led an important practice-changing clinical trial, protocol B-04, which showed that total mastectomy was just as effective as the more extensive Halsted operation [2]. This landmark study created the first major cracks in the theory that local aggressive treatment is “the” solution for BC and paved the way for future breast-conserving treatments. Indeed, NSABP’s protocol B-06, which was initiated in 1976 [3], showed that adding radiation therapy (RT) to the removal of the tumor alone (lumpectomy) was as effective as a mastectomy, while proving to be far less disfiguring.

This revolution in the surgical management of BC had also been initiated in Europe under the leadership of Umberto Veronesi, a visionary surgeon who ran a similar trial leading to the endorsement of lumpectomy with radiation as the gold standard “whenever possible” [4]. Large population studies in recent years supported this approach and even showed better long-term survival as a breast-conserving therapy (BCT) compared to mastectomy in women with early BC [5,6]. Veronesi also pioneered, together with Armando E Giuliano in the US, the next revolutionary step in surgery for BC, namely the sentinel node biopsy procedure (SNB), which can detect the sentinel node in the axilla, and thereby provide important staging information about the status of axillary nodes and spare patients without nodal involvement from axillary lymph node dissection (ALND) [7,8]. The Z0011 study showed that patients with a positive SNB can, in some clinical situations, be spared ALND without compromising recurrence rate, providing additional support to the diagnostic-staging importance of axillary evaluation while maintaining that it has no role in curing the disease. In 2017, the final results of the Z0011 study changed the common practice of performing ALND by showing that early BC patients with 1–2 sentinel lymph nodes containing metastases who were treated with sentinel lymph node dissection (SLND) alone had non-inferior overall survival compared with those treated with ALND [9].

SENTINA and ACOSOG Z1071 studies have shown that patients presenting with clinically node-positive disease, who are therefore candidates for primary systemic therapy, can be treated with SNB rather than ALND, however with relatively high false-negative rates [10,11]. Recent studies have also found ways to improve the accuracy of SNB with different kinds of LN marking [12], thereby adding another layer to this minimally invasive approach that we have seen in the last decade.

### 2.2. Radiation

Soon after Röntgen′s announcement of the discovery of the X-ray in 1895, and the discovery of radium in 1898 by Marie Curie, efforts were shifted to attempt to use radiation for the treatment of diseases [13]. Despite limited evidence of efficacy, the radiation of the chest wall and nodal drainage regions was adopted by many medical fields and institutions. The clear effectiveness of breast radiation was first provided by the previously described NSABP B-06 trial, which showed that adding RT to lumpectomy could provide a means to spare women to avoid mastectomy [3]. While the exact indications of post-mastectomy RT are still a matter of controversy, an important meta-analysis by the Early Breast Cancer Trialists Collaborative Group (EBCTCG) showed a 17% reduction in local recurrence and a 5% reduction in BC mortality in patients who had been radiated post-mastectomy [14]. Interestingly, in 1970, one of the NSABP’s first randomized trials failed to confirm improvement in survival for post-mastectomy radiation [15]; however, a large meta-analysis study in 2014 showed a reduction in local recurrence and BC mortality in patients with any number of involved axillary nodes post-mastectomy and axillary node dissection [16]. In the context of reconstruction, it is important to note that radiotherapy could damage the cosmetic results of reconstruction; therefore, the timing of radiation and the type of reconstruction should always be discussed by the radio-oncologist and the surgeon [17].

As part of the new paradigm, namely a shift from aggressiveness in local therapy to aggressiveness in systemic therapy, and given the emergence of new RT techniques, efforts focused on minimizing RT. New techniques, such as partial breast irradiation (PBI), intraoperative RT, or brachytherapy, have become available in the last twenty years. Despite the potential promise of these techniques, recent evidence suggests that intraoperative RT and brachytherapy may be less effective and more toxic in some situations, compared to conventional therapy [4,18]. On the other hand, large population studies have shown that partial-breast or reduced-dose radiotherapy and intensity-modulated radiotherapy techniques are as effective in the reduction in local relapse as whole breast irradiation post-lumpectomy in patients with early BC [19]. Moreover, partial-breast irradiation has shown better cosmetic results and less toxicity [20]. Caution is therefore warranted when implementing these strategies in general practice and it is crucial to select the right patients for these “softer” treatments.

The irradiation of the regional nodes (internal mammary, supraclavicular, and axillary) as part of the post-surgical treatment in node-positive or high-risk BC patients was a common practice until the 1980s when controversy arose between trials regarding the survival benefit, which was observed in preceding trials. Lymph node irradiation remained in use for post-mastectomy patients; however, the treatment’s benefit post-lumpectomy was unknown. Recent studies have shown that the addition of regional nodal irradiation to whole-breast irradiation improves disease-free survival (DFS) and reduces recurrence, but the influence on overall survival changes between the studies [21,22]. In 2020, a 15-year follow up of a large population study was published; however, researchers remain unable to demonstrate the benefit of regional lymph node irradiation on the overall survival [23].

## 3. Systemic Treatment

### 3.1. Hormonal Therapy

In 1895, George Beatson, a Scottish surgeon, showed that removing the ovaries from young women with BC leads to tumor shrinkage. The experiment that led to this discovery involved lactating rabbits. Beatson showed that proliferative tissue in the gestational breast degenerated into fatty tissue post-oophorectomy. This was a classic example of the dependency of breast tumors on the estrogen secreted by the ovaries. Soon, this strategy was adopted by many surgeons. To eliminate estrogen levels in BC patients, adrenalectomies and pituitary gland resections were suggested and performed as additive treatments.

The next evolution in hormonal therapy for BC was the discovery of nonsteroidal antiestrogens during the 1960s, as well as the identification of estrogen receptor (ER) by Elwood V Jensen [24]. Initially, the medical community exhibited a reluctance to pursue the development of these drugs as palliative treatment for BC and ignorance about the strong predictive role of ER for hormonal therapy effectiveness [25]. However, tamoxifen, a selective estrogen receptor modulator, showed dramatic responses and a relatively good safety profile in patients with metastatic BC [26]. These observations thus encouraged the NSABP to conduct a prospective randomized trial that evaluated tamoxifen in the adjuvant setting of patients with ER-positive BC in 1981. This landmark trial showed a 20% reduction in overall mortality for patients receiving tamoxifen for 5 years [27]. However, not all randomized studies with similar designs generated positive results; therefore, the level one evidence for the efficacy of adjuvant tamoxifen came later with the meta-analysis performed by the EBCTCG which firmly demonstrated a 31% reduction in BC mortality in tamoxifen-treated patients, with the benefit being confined to those patients with ER-positive tumors [28].

Future progress in hormonal treatment occurred when agents that systemically reduced estrogen levels by different mechanisms, i.e., aromatase inhibitors (AIs) in postmenopausal patients and luteinizing hormone-releasing hormones (LHRH) agonists in premenopausal patients, were shown to be highly effective in ER-positive BC [29,30]. Angela Brodie began developing the novel approach of targeting the enzyme aromatase to inhibit the synthesis of estrogen in the early 1970s and thus developed the drug formestane, the first AI to be used to treat BC [31]. The next generation of AIs used today was proven to be slightly more effective than the standard tamoxifen. Additionally and more importantly, sequencing both treatments and prolonging the treatment schedule appears to enable longer-lasting disease control [29,32].

The optimal duration of endocrine treatment (ET) remains uncertain and largely depends on the characteristics of the primary tumor, including nodal involvement. Recent studies show that recurrence rates after 5 years of ET decrease but remain significant [33]. The same studies also show that extending the duration of tamoxifen treatment for up to 10 years proved to further improve outcomes [34,35]. The benefit of extension of AI treatment beyond 5 years is still unclear even though numerous studies were motivated to determine it [36,37].

The importance of ovarian suppression in addition to the ET in premenopausal women with high-risk BC has been emphasized in recent years with the TEXT and SOFT trials showing higher DFSand overall survival (OS) in women treated with ET plus ovarian suppression. Better outcomes are observed of the combination of ovarian suppression with AIs rather than with tamoxifen. Longer follow-up also shows some benefit in 8-year freedom from distant recurrence [38]. In early 2022, a patient-level meta-analysis of 7030 women from four randomized trials comparing AI versus tamoxifen was published. The studies included in the meta-analysis followed premenopausal women with ER-positive BC receiving ovarian suppression who were treated with AIs or tamoxifen for 3 or 5 years. The results showed a reduced risk of BC recurrence using AIs for 5 years compared with tamoxifen in these patients. There was no significant difference between treatments for BC mortality and overall mortality [39].

In the case of progression during treatment or relapse, the addition of a second therapeutic agent is required to enhance the efficacy of AIs. One of the most common additions is CDK4/6 inhibitors. In the early 1980s, Paul Nurse discovered the cyclin-dependent kinase (CDK) genes, which are regulators of the cell division cycle [40]. Later research showed that some BC tumor cells overexpress cyclin D1 and the dysregulation of the cyclin D1:CDK4/6 axis has a role in BC [41]. Therefore, the inhibition of CDK4/6 may inhibit generations of BC cells. Initial basic science studies did not show promising results; however, 10 years ago, studies began to discover the in vitro role of CDK4/6 in BC cells. In 2016, the PALOMA-2 trial was published, the first phase 3 clinical trial to confirm the efficacy of Palbociclib (CDK4/6 inhibitor) combined with Letrozole (an AI) for women with ER-positive, HER2-negative advanced BC. It showed significantly longer progression-free survival (PFS) when compared with only letrozole therapy [42]. Ribociclib was also investigated in advanced BC patients in three different randomized trials: MONALEESA-2, MONALEESA-3, and MONALEESA-7. MONALEESA-2 tested ribociclib plus letrozole in postmenopausal patients with HR+, HER2− advanced BC. MONALEESA-3 tested ribociclib plus fulvestrant in a similar population. Both MONALEESA-2 and MONALEESA-3 showed improved PFS with ribociclib plus ET versus placebo plus ET [43,44]. MONALEESA-7 examined ribociclib or placebo plus ET (AI or tamoxifen) in premenopausal and perimenopausal women and showed significantly longer PFS and overall survival in the ribociclib arm [45]. Following these results, the addition of CDK4/6 to the hormonal treatment of women with early BC was also studied. In 2020, Abemacilclib (CDK4/6 inhibitor) plus adjuvant ET was shown to improve DFS more than ET alone in high-risk patients with HR+, HER2-node-positive early BC [46]. Palbociclib, however, failed to show the same results in recent studies [47].

An additional attack target in BC treatment is the phosphatidylinositol 3-kinase (PI3K)/protein kinase B (AKT)/mammalian target of rapamycin (mTOR) signaling pathway. Aberrant signaling through this pathway causes resistance to anti-estrogen therapy in ER-positive BC [48]. As a result, Everolimus (an mTOR inhibitor) was examined in several trials over the past decade in combination with numerous anti-estrogen therapies, in advanced BC, HR-positive patients who were previously treated with AI. The BOLERO-2 trial published in 2012 demonstrated that treatment with Everolimus combined with Exemestane more than doubled PFS compared with a placebo alternative. Furthermore, combining Everolimus and Tamoxifen showed an increased clinical benefit rate and overall survival in a phase 2 study [49]. A consequent phase II study explored the efficacy of Everolimus with Fulvestrant (ER antagonist), showing doubled PFS compared with Fulvestrant alone [50].

PIK3CA mutations are detected in over 40 percent of hormone receptor-positive BC. The SOLAR-1 trial tested treatment with Alpelisib (PI3Kα-specific inhibitor) with Fulvestrant in PIK3CA mutation, HR-positive advanced BC patients. This combination almost doubled PFS [51]. In the neoadjuvant setting, the combination of alpelisib and ET neo-adjuvantly was examined in the NEO-ORB trial in postmenopausal women with HR+, HER2− early BC, but in contrast to the previous results in advanced BC patients, the addition of alpelisib did not improve objective response rate or pathologic complete response compared with placebo [52]. The therapeutic molecular targets in BC are summarized in Figure 1.

### 3.2. Chemotherapy

The premise that BC has a strong tendency towards early dissemination became widely recognized during the mid-1950s; however, it was the development of chemical weapons (nitrogen mustard) during World Wars I and II that led to the establishment of national drug development efforts [53]. The rationale for adjuvant chemotherapy came from laboratory findings that showed that systemic agents given after surgeries improved outcomes, probably by destroying micro-metastases.

At first, during the 1980s, a series of studies tested the concept that adjuvant chemotherapy could have a positive effect on survival. Gianni Bonadonna was instrumental in validating the concept that polychemotherapy (cyclophosphamide, methotrexate, and fluorouracil) as adjuvant treatment would improve outcomes [54]. Nevertheless, many other studies lacked statistical power and were negative. Once more, the role of the EBCTCG was instrumental in that their meta-analysis of 194 studies conducted between 1985 and 2000 showed a 20–38% reduction in BC mortality depending on the age of the patient, and a possible additive value for anthracylines [28]. Thus, at the beginning of the 21st century, a second wave of randomized trials showed that the addition of taxanes to a fixed anthracycline-based control regimen further reduced BC mortality by an absolute 3% at 8 years post-treatment [55].

Since adjuvant chemotherapy proved itself to be very effective, it seemed rational to evaluate whether initiating chemotherapy at diagnosis, when the tumor burden is the lowest, would improve the outcome. In fact, in the 1970s and early 1980s, Gabriel Hortobagyi and Claude Jacquillat had already initiated studies involving neoadjuvant chemotherapy for BC patients and showed a reduction of approximately 50% in tumor size [56,57]. The NSABP B18 study demonstrated that neoadjuvant therapy reduced tumor size and increased the number of women able to undergo lumpectomy; however, there was no shown impact on overall survival [58]. Another randomized trial performed by the European Organization for Research and Treatment of Cancer (EORTC) also failed to demonstrate a survival advantage [59]. Despite the findings that survival is not improved by neoadjuvant chemotherapy compared with adjuvant chemotherapy, a few advantages associated with neoadjuvant treatment were demonstrated, including a significant response to chemotherapy, an increase in the potential to offer breast-conserving treatment, and the ability to assess the response of the primary tumor to a particular chemotherapy regimen.

Currently, the usual regimen of chemotherapy for BC is anthracyclines and taxanes, in the adjuvant and neoadjuvant settings. Ten years ago, it was discovered that triple-negative BC are sensitive to platinum-based chemotherapy [60]. A few phase-2 and -3 trials published in recent years confirmed this evidence, showing that the addition of carboplatin to polychemotherapy neoadjuvant treatment increases the rate of pathological complete response in triple-negative BC patients [61,62]. Further research, however, is still needed.

Progress in chemotherapy treatments continued as trials began to investigate the efficacy of capecitabine, a prodrug of fluorouracil, in BC patients. The very first evidence for its efficacy was in 1999 when a phase-2 trial showed the effectiveness of capecitabine treatment in patients with taxane-refractory metastatic BC [63]. Further studies supported these results in advanced BC patients and later on, the treatment was tested in patients with early BC. A large phase-3 trial published in 2017 showed prolonged DFS and OS in patients with HER2-negative residual invasive BC after neoadjuvant chemotherapy who were treated adjuvantly with standard postsurgical treatment plus capecitabine [64]. Although today this treatment is an approved option in selected patients, there remain inconsistent results in different studies. Consecutively, the recently published CBCSG010 trial tested the addition of capecitabine to standard adjuvant chemotherapy in TNBC and showed significant improvement in DFS [65]. The cell cycle targets of chemotherapy and novel drugs are summarized in Figure 2.

### 3.3. Biological Therapy

In 1987, Dennis Slamon reported that the HER2/neu oncogene encoding for a member of the epidermal growth factor family was amplified in about 20% of BC patients and was associated with a shorter time to relapse and a lower survival rate [28]. Scientists at Genentech characterized monoclonal antibodies that are reactive to the HER2/neu gene product and are therefore able to inhibit the growth of cancer cells that express it. In 1991, trastuzumab (Herceptin^®^) was selected as the first monoclonal antibody to be tested in human subjects and, in 1998, it received FDA approval for the treatment of HER2-positive metastatic BC based on a pivotal trial in which it was combined with paclitaxel [29].

At the turn of the 21st century, several randomized prospective clinical trials were launched to evaluate the role of trastuzumab when given in addition to chemotherapy in the adjuvant setting [30,31,32,33]. These generated highly consistent results showing that the addition of trastuzumab to chemotherapy decreased recurrence by 50% and mortality by 30%.

In the last decade, there has been an attempt to evaluate other drugs that target HER2. Lapatinib, which is a tyrosine kinase inhibitor, and pertuzumab, which is a monoclonal antibody against the dimerization domain of HER2, have both been combined with trastuzumab and chemotherapy and demonstrated improved efficacy in advanced disease and the neoadjuvant setting [34,35]. Results in the adjuvant setting arrived in 2017 in a study that tested the addition of pertuzumab to standard adjuvant chemotherapy in addition to 1 year of trastuzumab and showed a small improvement in DFS [66].

Despite the impressive increase in OS in patients treated with trastuzumab, there is a portion of patients who do not respond initially or relapse soon after the end of the treatment. Therefore, the necessity to investigate treatments that have improved response and efficacy developed. A decade ago, the idea of using antibodies to deliver toxic agents to cancer cells began to rise. As a result, in 2007, the first in vitro and in vivo evidence of trastuzumab-DM1 (T-DM1—microtubule-depolymerizing agent) conjugates was published, showing the greater activity of T-DM1 compared with unconjugated trastuzumab in HER2-overexpressing tumor cells [67]. The first phase 3 randomized trial was published in 2012, comparing T-DM1 with lapatinib plus capecitabine in HER2-positive advanced BC patients previously treated with trastuzumab and a taxane, showing prolonged OS and less toxicity in the T-DM1 group [68]. More supporting evidence was published in further years, but the information regarding T-DM1 treatment in early BC arrived in the last couple of years, with the results of the KATHERINE trial that tested the efficacy of adjuvant T-DM1 in HER2-positive early BC patients with residual invasive cancer at surgery after completion of standard neoadjuvant therapy showing a 50% lower risk for recurrence with adjuvant T-DM1 than with trastuzumab [69]. A similar antibody-drug conjugate-trastuzumab-deruxtecan has been used in cases of advanced HER2-positive BC. It is composed of an anti-HER2 antibody conjugated to a cleavable tetrapeptide-based linker as well as a topoisomerase I inhibitor. One of a multitude of studies to test the safety and efficacy of trastuzumab-deruxtecan in HER2-positive advanced BC patients who were previously treated with T-DM1 showed a manageable safety profile and initial evidence of activity [70]. Additionally, the phase-2 DESTINY trial examined this treatment in a population with similar baseline characteristics and confirmed that most responded to therapy in different response durations [71].

In recent years, the antibody-drug conjugate sacituzumab govitecan was investigated as a targeted therapy for triple negative BC. It comprises an antitrophoblast cell-surface antigen 2 (Trop-2) humanized IgG1κ antibody coupled to a small molecule topoisomerase inhibitor (SN-38). Many epithelial tumors (including triple negative BC) overexpress Trop-2; thus, sacituzumab govitecan enables targeted delivery of SN-38 to the tumor. The first evidence of efficacy in triple negative BC patients was published in 2017 in the IMMU-132 trial, a phase 1/2 trial that followed various types of advanced solid cancer patients. A total of 69 of the participants were metastatic triple-negative BC patients refractory or relapsed after one or more standard line of therapy, who received 21 days of treatment with sacituzumab govitecan. The results showed a 30% objective response rate [72]. In 2021, the ASCENT trial, a large phase-3 randomized trial, showed significantly prolonged PFS and OS in advanced triple-negative BC patients who received sacituzumab govitecan compared to single-agent chemotherapy [73]. Though promising, such treatment is yet to be investigated in early BC patients.

The BRCA1/2 mutation, which predisposes some women to develop the disease, is an additional current challenge in the treatment of triple-negative BC [74]. These types of cancer are characterized by their deficiency in homologous recombination repair [75], which led researchers to investigate the efficacy of PARP (poly (adenosine diphosphate-ribose) polymerase) inhibitors in the treatment of BC patients with BRCA1/2 mutations. PARP inhibitors obstruct the repair of DNA single-strand breaks, thus allowing such transcription errors to accumulate and lead to tumor-cell death. This therapy’s principle can therefore be utilized in these patients to selectively kill those cells with homologous recombination repair deficiency, i.e., cells with a BRCA1/2 mutation. Initial evidence regarding the efficacy and safety of the treatment’s benefit was shown in the OlympiAD trial, which compared treatment with Olaparib with standard therapy in patients with HER2-negative metastatic BC and a germline BRCA mutation after one or two previous chemotherapy regimens. Results showed significantly longer PFSin the Olaparib group [76]. Similar results were also shown in the EMBRACA trial where the use of Talazoparib was compared with standard therapy in advanced BC and germline BRCA1/2 mutation patients [77]. The OlympiA trial tested the use of olaparib as first-line adjuvant therapy in patients with BRCA1/2 mutations early BC and high risk of recurrence and showed longer invasive-DFS when compared with placebo [78].

## 4. Conclusions

The last decades have revolutionized BC oncology and changed it from an empirical domain to one that is increasingly evidence-based. While local procedures, such as surgery and RT, are becoming less invasive and more accurate, the systemic part of the treatment is becoming more complex. In the last decade, medical professionals have witnessed a trend and movement towards specific targeted therapies, which harbor less systemic effect on healthy cells. These new treatments and the published trials that explore them demonstrate the wider understanding of the molecular and biological characteristics of the tumors and the scientifically savvy ways of using them to our advantage in treatment. Today, a patient with BC may find herself confronted with multiple different options for post-surgical systemic treatments: anthracyline-taxane backbone chemotherapy, capecitabine, hormonal therapy (that might change from tamoxifen to an AI and extend to 10 years), the option of ovarian suppression, trastuzumab, pertuzumab for one year or T-DM-1 and more. The side effects and emotional distress from such multi-pharmaceutical treatments are troublesome, not to mention their economic burden. The benefit of each drug and the number of patients who need to be treated to cure one patient in such a scenario are not clear. Strong leadership is essential from the designers of clinical trials, cancer translational scientists, and the medical oncologists who treat patients with BC to find more efficient ways to tailor adjuvant regimens to the needs of individual patients.

## Figures and Tables

**Figure 1 genes-13-00960-f001:**
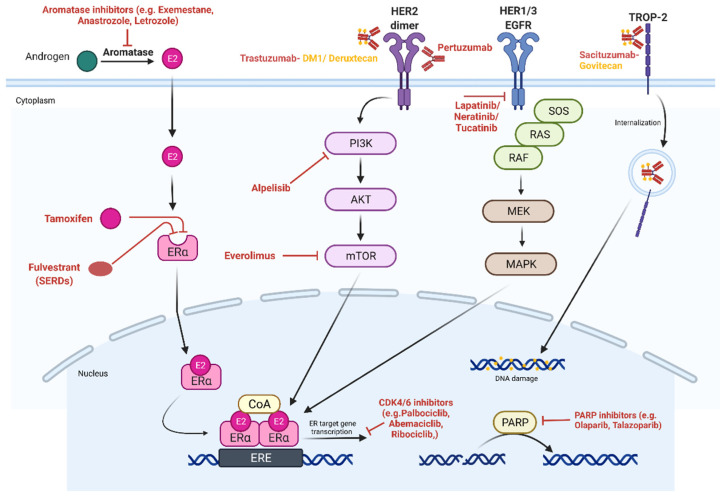
Therapeutic molecular targets in breast cancer. ER (estrogen receptor), SERD (selective estrogen receptor degrader), HER1/2 (human epidermal growth factor receptor 1/2), EGFR (epidermal growth factor receptor), and TROP-2 (trophoblast cell surface antigen 2).

**Figure 2 genes-13-00960-f002:**
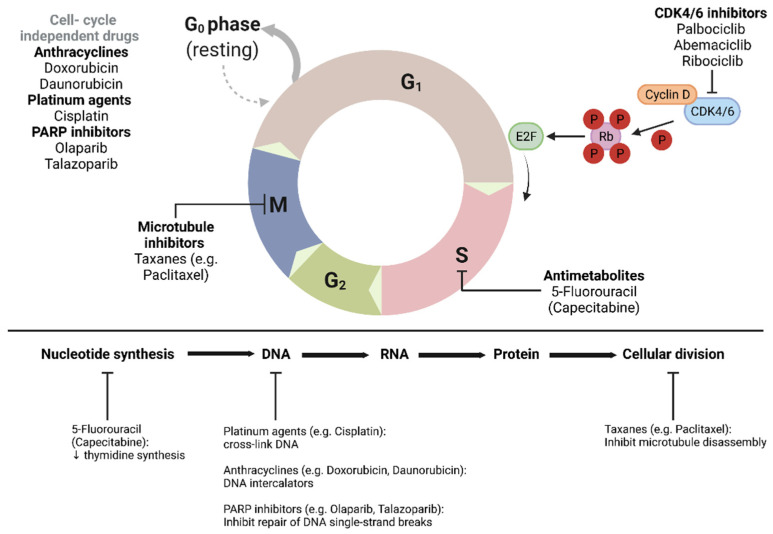
Cell cycle targets of chemotherapy and novel drugs.

## Data Availability

Not applicable.

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
