# Peer review of "The History of Early Breast Cancer Treatment"

_genes, 2022, doi:10.3390/genes13060960_

Round 1
Reviewer 1 Report
This is a comprehensive work and overview on the history of breast cancer treatment.
Some passages could be improved, see below. It also needs some editing (typos etc.).
- 110: Sir George Thomas Beatson: Maybe mention that he first discovered the anti-hormone effect of an oophorectomy in rabbits, when he observed in lactating rabbits a stop in milk production after the surgery.
- 144: You should also cite the ATTOM trial here.
- 152: Please also mention the latest meta-analysis of the EBCTCG on this
Aromatase inhibitors versus tamoxifen in premenopausal women with oestrogen receptor-positive early-stage breast cancer treated with ovarian suppression: a patient-level meta-analysis of 7030 women from four randomised trials.
- 161-168: Why do you not mention Ribociclib? To date, it is the only CDK4/6 inhibitor that has prolonged overall survival significantly in a total of three randomized trials (MonaLeesa-2/3/7), it deserves to be mentioned here.
- 186: you could mention the NEO-ORB trial here.
- Figure 1: Please re-name Aromasin to exemestane, as you mention anastrozole and letrozole, which both are substance names, too. There is a typo in Fulvestrant (SERD instead of SRED). A legend for the abbreviations is missing. If you mention all of the approved substances in this figure, you should mention ribociclib, too. Is there also a way to include sacituzumab-govitecan in this figure?
- Figure 2: Typo in Talazoparib and Capecitabine. Again, include Ribociclib, please.
- 283: It is unusual to use the trade name (Enhertu) instead of the substance name in a publication.
- 291: predisposes some „women“ to develop the disease, not breast cancer patients yet.
- A passage on sacituzumab-govitecan would be nice, too.
Reviewer 2 Report
This is an excellent review of the development of the treatment of breast cancer. It should be published ASAP as the treatment of breast cancer continues to evolve and if delayed the manuscript will have to be updated/
Author Response
Thank you for your comment